# A high-resolution reanalysis of global fire weather from 1979 to 2018 – Overwintering the Drought Code

Megan McElhinny[1,2], Justin F. Beckers[2], Chelene Hanes[3], Mike Flannigan[4], Piyush Jain[2]

[1]Environmental Sciences, University of British Columbia
[2]Natural Resources Canada, Canadian Forest Service, Northern Forestry Centre, 5320 – 122nd Street, Edmonton, AB T5H 3S5, Canada.
[3]Natural Resources Canada, Canadian Forest Service, Great Lakes Forestry Centre, 1219 Queen St. E, Sault Ste. Marie, ON P6A 2E5, Canada
[4]Department of Renewable Resources, University of Alberta, Edmonton, AB T6G 2H1, Canada.

*Correspondence to*: Megan McElhinny (megan.e.mcelhinny@gmail.com)

**Abstract**

We present a global high-resolution calculation of the Canadian Fire Weather Index (FWI) System Indices using surface meteorology from the ERA5-HRS reanalysis for 1979-2018. ERA5-HRS represents an improved dataset compared to several other reanalyses in terms of accuracy, as well as spatial and temporal coverage. The FWI calculation is performed using two different procedures for setting the start-up value of the Drought Code (DC) at the beginning of the fire season. The first procedure, which accounts for the effects of inter-seasonal drought, overwinters

the DC by adjusting the fall DC value with a fraction of accumulated overwinter precipitation. The second procedure sets the DC to its default start-up value (i.e. 15) at the start of each fire season. We validate the FWI values over Canada using station observations from Environment and Climate Change Canada and find generally good agreement (mean Spearman correlation of 0.77). We also show that significant differences in early season DC and FWI values can occur when the FWI System calculation is started using the overwintered versus default DC values, as is

highlighted by an example from 2016 over North America. The FWI System moisture codes and fire behavior indices are made available for both versions of the calculation at https://doi.org/10.5281/zenodo.3626193 (McElhinny et al., 2020), although we recommend using codes and indices calculated with the overwintered DC, unless specific research requirements dictate otherwise.

## 1 Introduction

Climate reanalyses provide a numerical and geospatial description of past and present climate (Bengtsson *et al.,* 2007). This method of climate simulation assimilates weather observations into dynamic climate models of the atmosphere and relevant Earth systems to represent the atmospheric and surface states at a given time, usually for a historical period of multiple decades to near-present. The gridded product of reanalysis is spatially and temporally continuous

for the duration of the model simulation, and has the added benefit of producing data in remote areas that are sometimes inaccessible to direct monitoring (Bengtsson *et al.,* 2007). The best climate reanalyses use the same model



configuration for the duration of the simulation, thus eliminating inhomogeneities that may occur through other modes of climate tracking, and providing a useful tool for studying weather-related phenomenon.

Past research in the field of reanalysis and fire weather has analysed the correlation between metrics of fire danger produced by reanalyses and those produced from weather stations at local to continental scales. In comparing observed and reanalysis-derived indices of fire weather, reanalyses have been found to be an effective tool for indicating fire danger (Bedia *et al.,* 2012; Venalainen *et al.,* 2014; Field *et al.,* 2015). Other studies have investigated the relationship between fire weather indices calculated from reanalyses and measures of the fire regime, such as annual area burned (Bedia *et al.,* 2014), trends in fire season length (Jain *et al.,* 2017), and quantification of global

seasonal fire danger (Vitolo *et al.,* 2019). Reanalyses have also been used to investigate the spatio-temporal variation of fire danger indices across continents (Lu *et al*., 2011) and to develop new indices that investigate the validity of incorporating synoptic and meso-scale weather processes into metrics of fire behavior (Srock *et al.,* 2018). Recently, research has begun to investigate the application of reanalyses in prediction of future fire weather and fire behavior patterns by evaluating how they can supplement the coarse resolution of Global Climate Models on local scales

through statistical downscaling (Bedia *et al.,* 2013). Although climate reanalysis has been found to be a useful and reliable tool for calculating indices of fire behavior, some metrics of fire danger require specific temporal weather measurements, such as noon local standard time measurements, that many reanalyses cannot directly provide (Herrera *et al.,* 2013). However, the concerns raised around this shortcoming have been addressed in recent years by new reanalysis products with better temporal resolution, among other improvements.

Many countries, including Canada, use the Canadian Fire Weather Index (FWI) System to determine the effects of weather on forest fuel moisture and subsequently fire behaviour (Lawson and Armitage, 2008). The FWI System considers surface temperature, relative humidity, 24-hour accumulated precipitation, and wind speed at 10m to calculate moisture in three fuel layers respectively represented by three moisture codes: the Fine Fuel Moisture Code (FFMC), the Duff Moisture Code (DMC) and the Drought Code (DC). These values, plus wind speed, are then

used to calculate indices of potential fire behaviour; Initial Spread Index (ISI) and Build-up Index (BUI) from which the Fire Weather Index (FWI) and Daily Severity Rating (DSR) are produced.

The DC is one of three moisture codes that impacts fire behaviour and is the metric that tracks moisture in the deepest layer of forest floor fuels as well as in large, dead woody debris (Wotton, 2009). Due to its depth, the DC is the slowest changing moisture code with a time lag of 52 days (Van Wagner, 1987). Essentially, the DC value

decreases with effective rainfall and increases with evapotranspiration so that higher values indicate a higher likelihood that a wildfire will persist and smoulder (Van Wagner, 1987).

In areas where winter precipitation is sufficient (i.e. greater than 200mm rain or snow equivalent), moisture reserves are typically recharged in the spring so that the default DC value of 15 represents near saturation of deep organic layers (Alexander, 1982; Lawson and Armitage, 2008). However, when this is not the case, an alternative

method to start-up the FWI calculation uses an overwintered value of DC. This value is determined from the final DC of the preceding fire season, representing any potential fall moisture deficit, and a percentage of overwinter precipitation, assumed available to recharge that deficit (Lawson and Armitage, 2008). The main body of thought behind using the overwintered DC is that it accounts for fall drought conditions and/or dry winter conditions, and thus



may indicate drier moisture conditions leading to more severe fire weather risk at the beginning of the fire season than

is suggested by the default DC.

A number of empirical field studies further support the need for overwintering the DC when calculating FWI System Indices.  Lawson and Dalrymple (1996) describe a method for ground-truthing (i.e. the process of calibrating from and/or validating against field sample measurements) the DC with destructive sampling, which can be executed at any time during the fire season. They concluded that overwintering the DC was adequate for broad areas but that

site-specific calibration may still be necessary. Confirming this finding, Bourgeau-Chavez et al. (2007) demonstrated that the default start-up value of DC was not sufficient for describing spring fuel moisture in Alaska and Girardin et al. (2006) showed that both area burned and number of large fires were correlated with previous season's DC. Furthermore, Wilmore (2001) showed that default DC values overpredicted spring fuel moisture and that overwintering the DC led to improvements in estimates of drought conditions, although this could be further improved

upon using an alternative site-specific overwintering equation.

Although these papers largely indicate that the overwintered DC is more representative of actual conditions than the default DC, many note that regional adjustments for the carry-over fraction from the previous season's fall moisture and the coefficient for effectiveness of winter precipitation in recharging moisture reserves in the spring is necessary when calculating the overwintered DC (Lawson and Armitage, 2008; Anderson and Otway, 2003). In

analysing the conditions leading up to the Fort McMurray Wildfire in Alberta, Canada, it was found that accounting for observations of fuel moisture when calculating the start-up DC value can additionally improve the accuracy of fire danger detection (Elmes *et al.,* 2018).

ERA5-HRS is a high resolution reanalysis dataset named for and produced by the European Centre for Medium-Range Weather Forecasts (ECMWF) ReAnalysis High ReSolution product, of which the dataset is the fifth

generation (after FGGE, ERA-15, ERA-40, and ERA-Interim) (Hennerman and Berrisford, 2019). The spatial and temporal continuity and resolution of this dataset makes it a useful tool for analysing past weather and associated phenomenon. The main purpose of this paper is to document the calculation of FWI System indices using the global ERA5-HRS reanalysis (hereafter known as ERA5). We perform the calculation using both default and overwintered DC start-up values, the latter being important for some regions with snow cover or ground freeze over winter.


## 2 Data

The ERA5 reanalysis product is produced from the CY41R2 global ensemble system of the ECMWF Integrated Forecast System (*Copernicus Climate Change Service (C3S) (2017)).* Weather observations from satellites and in-situ data from the World Meteorological Organization are integrated into the global ensemble using 4-dimensional

variational analysis data assimilation (Hennerman and Berrisford, 2019).

The high resolution realisation is 31km globally or 0.28125 degrees on a reduced Gaussian grid (output at 0.25 degrees on a regular geographic grid), which provides an improvement in precision over its predecessor, ERA-Interim, for which the resolution was 79km globally (Hennerman and Berrisford, 2019). Additionally, ERA5 has a finer resolution compared to other global reanalysis products including the NCEP North American Regional

Reanalysis (NARR), NCEP-DOE Global Reanalysis 2, and NASA's Modern Era-Retrospective Analysis for Research



and Applications (MERRA) as well as MERRA-2. The ERA5 dataset covers 1979 to 2-3 months before present (our calculation only used data up to 2018 to obtain a full final year), on an hourly scale producing numerous global climatological variables including surface and upper atmosphere quantities. For this study we obtained surface variables including temperature (K), dewpoint temperature (K), U and V-components of wind (m/s), precipitation (m),

and   a   land-sea   mask,   available   from   the   ECMWF   Climate   Data   Store   (CDS)   website (https://cds.climate.copernicus.eu/cdsapp#!/dataset/reanalysis-era5-single-levels?tab=form, accessed 17 May 2019).

Compared with the ERA-Interim product, ERA5 has been shown to perform better with respect to variation in data quality over space and time, tropospheric representation, representation of tropical cyclones, soil moisture accuracy, sea surface temperature and sea ice cover detection, the global precipitation and evaporation balance, and

precipitation over land especially in the deep tropics (Hennerman and Guillory, 2019). One of the most impressive improvements to the ERA5 dataset is with respect to precipitation modelling. Multiple studies have been conducted to determine the accuracy with which ERA5 detects various aspects of past precipitation events. One study, conducted in the Assiniboine River Basin of the Northern Great Plains, found that of six reanalysis products, including CaPA, ERA-Interim, ERA5, JRA-55, MERRA-2, and NLDAS-2, the ERA5 dataset consistently performed in the top three

regarding precipitation detection, correlation to observed precipitation events, Mean Absolute Error, and Root Mean Square Error (RMSE) (Xu *et al.,* 2019). A study in the Fram Strait, off the coast of Greenland, found that of ERA5, ERA-Interim, JRA-55, CFSv2, and MERRA-2, the ERA5 reanalysis produced the most accurate simulation of radiosonde profiles over the Strait and showed the lowest vertically averaged absolute biases for every variable except relative humidity, which was best simulated by JRA-55 (Graham *et al.,* 2019). Over North America, ERA5 was

consistently found to have lower precipitation (and temperature) biases than ERA-Interim, reducing the median gap between observations by 40% compared to its predecessor (Tarek *et al.,* 2019). ERA5 performance was found to be equivalent to directly observed data for most of the region, excluding the Eastern U.S. where observations were more accurate. Finally, a study conducted over the contiguous United States found that of 26 (sub-)daily precipitation datasets, ERA5 had the best performance of the compared uncorrected precipitation products, meaning those using

only satellite and/or reanalysis data (Beck *et al.,* 2019). The significant capabilities of the ERA5 reanalysis product indicate the impressive advances in earth system modelling that have been made in recent years by the ECMWF. With the improved representation of surface weather from ERA5 in mind, we used the product to calculate indices of fire weather for the entire globe.

**3 Methods**

**3.1 Fire Weather Input Variables**

The requirements of the FWI calculation stipulate that 2m temperature (°C), 10m wind speed (km/hr), and 2m relative humidity (%) measurements be taken at noon LST for each global time zone and that precipitation (mm) be accumulated over a 24-hour period between noon LST of each day (Van Wagner, 1987). See Lawson and Armitage (2008) for details on weather station sensors. To facilitate subsetting the data to noon for each time zone, we created

a Land Index from the ECMWF land-sea mask and a shapefile of global time zones (retrieved from the Natural

Earth website, https://www.naturalearthdata.com/downloads/10m-cultural-vectors/timezones/, accessed 18 May, 2019). The Land Index contained information regarding the location of each ECMWF grid cell over land or water as well as the time zone (as an offset from UTC±00:00) the grid cell covered. It should be noted that cells containing any amount of land were considered to be completely over land, and that cells entirely over water were not

processed in subsequent steps of the project.

To account for the noon LST requirement, the UTC offset values contained in each grid cell of the Land Index were used to select the first noon LST layer of each ECMWF monthly surface weather variable. A sequence of 24-hour increments was then applied to each cell, starting at the first noon LST layer, to select (or sum in the case of precipitation) 24-hour increments between noon LST of each day.

To account for the correct FWI units, the temperature (K) and accumulated precipitation (m) data were converted to °C and mm respectively. Relative humidity was calculated from the subsetted datasets for temperature and dewpoint temperature according to a derivation of the Rothfusz regression (Rothfusz, 1990) Eq (1) and (2):

$$a = \frac{112.0 - 0.1T + Td}{112.0 + 09.T} \tag{1}$$

$$RH = 100a^8 \tag{2}$$

where T is temperature (°C), $T_d$ is dewpoint temperature (°C), and RH is relative humidity (%). For wind speed, the subsetted datasets for the U-component and V-component of wind were converted to wind speed according to Eq (3):

$$WS = 3.6\sqrt{u^2 + v^2} \tag{3}$$

where u is the U-component of wind (m/s), v is the V-component of wind (m/s), and WS is wind speed (km/hr). Once each weather variable was subset into daily noon LST values and converted into the correct units, all monthly datasets

for a year were bound together to produce annual datasets of daily values for each element of fire weather.

**3.2 Overwintering Masks**

**3.2.1 Overwintered Drought Code**

In regions covered by snow over winter, the fire season is considered to be active on the third day after snow has

disappeared and the fire season is considered to be over when snow covers the ground. Alternatively, as a proxy to the snow condition, the fire season is considered to be active on the fourth day following three consecutive days with a maximum temperature of 12°C or higher, and the fire season is considered to be over after three consecutive days with maximum temperature of 5°C or lower (Wotton and Flannigan, 1993; Lawson and Armitage, 2008). Using this definition of overwintering means that the fire season can switch on and off throughout the year. For example, the

upper and lower maximum temperature thresholds may be met multiple times throughout the year resulting in short periods of fire season in the shoulder seasons (where the maximum daily temperature condition is met for short periods before and/or after the main fire season), in addition to a longer fire season period. The 24-hour maximum temperature of each day (using local standard time) were calculated from the ECMWF data using the hourly 2m temperature data. Accounting for the maximum temperature thresholds, we created binary masks of overwintering for each year from

the annual datasets of midnight LST 24-hour maximum temperature, with the process of overwintering the DC in mind.



### 3.2.2 Default Drought Code

When the default DC is used to start-up the FWI calculation, it is not desirable to include short shoulder fire seasons since the DC value is reset at the beginning of each fire season periods, which leads to discontinuities in the calculated

codes and indices. Thus, we only used the longest continuous fire season period from the Drought Code Overwintering Masks, by creating a mask for default DC with a single fire season start and end date per year in each grid cell of the Northern Hemisphere (NH) and Southern Hemisphere (SH). For NH grid cells, we saved the longest continuous fire season between January 1 and December 31 of the same calendar year and for SH cells, we saved the longest fire season between July 1 of one calendar year and June 30 of the next calendar year, with each time period

chosen to contain the boreal and astral summers respectively. Although SH cells were processed across years, the final product was organized to contain the overwintering status values for each cell on the globe for a calendar year in both the NH and SH. It is important to note that in some cases, a fire season run would extend beyond the defined end of year. Here, we added the length of days for which the fire season extended into the new year to the length of the fire season in the preceding year. Nevertheless, the main fire season was still defined as the longest run of fire season days

within the defined year (corresponding to each hemisphere) for each calendar year.

### 3.3 Annual FWI System Indices

Daily FWI System outputs, including FFMC, DMC, DC, ISI, BUI, FWI, and DSR were calculated using the cffdrs package in R (Version 1.8.5, Wang *et al.,* 2017) with the elements of fire weather data as inputs. When accounting for the overwintered DC, we programmed the fwiRaster function according to a Delta mask. The Delta value for each day

was calculated by subtracting the previous day's overwintered DC mask value from current day's mask value for each grid cell. This created four cases:

Case 1. The Delta mask was equivalent to 1. This indicated that it was the first day of overwintering (the fire season was inactive) at that location and thus we saved the DC of the previous day and the 24-hour accumulated precipitation of the current day.

Case 2. The Delta mask was equivalent to 0 and the current day's overwintering mask was equivalent to 1. This indicated that the overwintering status of the location was active (but it was not the first day of overwintering) and thus we saved the sum of the current day's precipitation and all precipitation since overwintering began.

Case 3. The Delta mask was equivalent to -1. This indicated that it was the first day of the fire season at that location and thus we calculated the start-up DC (a.k.a. the overwintered DC) from the saved DC value when

overwintering began and the precipitation that accumulated through the overwintering period, using the overwintering drought code function of the cffdrs package (Wang *et al.,* 2017). The final value of accumulated precipitation represents the total value of precipitation that fell during the period defined by the maximum temperature threshold criteria. Additionally, we set the FFMC and DMC to the default values of 85 and 6 respectively and stopped accumulating precipitation.

Case 4. The Delta mask was equivalent to 0 and the current day's overwintering mask was equivalent to 0. This indicated that the fire season status of the location was active (but it was not the first day of the fire season) and



thus the FWI calculation was reliant on the current day's weather variables and the previous day's FWI moisture code outputs.

When accounting for the default DC, the Delta mask was produced from the Default Drought Code Overwintering Masks as above. However, this resulted in only two relevant cases; Case 3 and 4. Case 4 was the same for the overwintered DC situation, but Case 3 was different in that the start-up DC was set to 15, rather than calculating the overwintered DC value.

In the case of the overwintered DC, the adjusted start-up values of DC were calculated using the wDC function in the cffdrs R package. In particular, we set the two required coefficients for this function as a = 1 (representing carry-over fraction of last fall's moisture) and b = 0.75 (default value of effectiveness of winter precipitation in recharging moisture reserves in spring). As noted by Lawson and Armitage (2008) and Anderson and Otway (2003), the overwintered DC is most accurately represented when regional conditions are analysed and the coefficients of the wDC function are adjusted accordingly. However, the ERA5 dataset did not contain information that allowed us to vary these coefficients and thus we chose the default values.

FWI Indices in the overwintered and default DC situations were calculated for Case 3 and 4. When Case 3 was identified, FWI Indices were calculated from the default FFMC and DMC, that day's values for the elements of fire weather, and the overwintered DC or default DC value depending on which overwintering mask was used. When Case 4 was identified, FWI indices were calculated from that day's elements of fire weather and the previous day's moisture codes in both Drought Code situations. We produced two final datasets of daily FWI System indices for 1979 to 2018: the first used the overwintered DC value at fire season start-up and calculated FWI values each time the maximum temperature thresholds were met, and the second used the default DC value at fire season start-up and only produced FWI values for the longest annual fire season in each hemisphere.

## 4 Analysis

### 4.1 Climatologies

Mean FWI values vary spatially and temporally based on climatological conditions and surface topography. Figure 1 shows monthly climatologies of mean FWI values for January, April, July, and October, which are indicative of global seasonal changes in FWI values. Note that the absence of values in the Northern latitudes for January, April, and October represents grid cells where the FWI System calculation is suspended because it is outside of the fire season period.
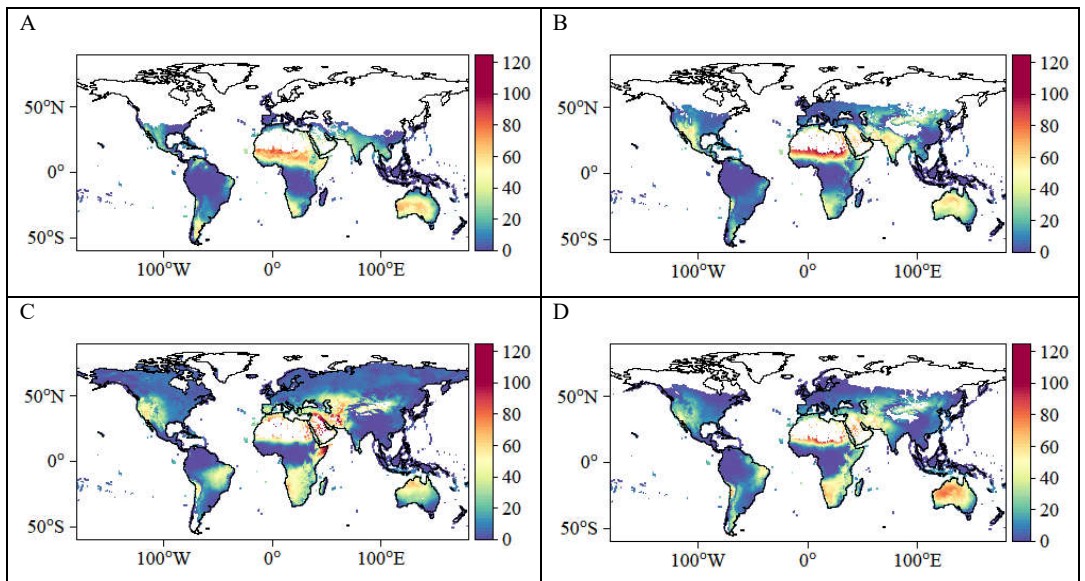

Fig. 1: Monthly climatologies of mean FWI values for January (Panel A), April (Panel B), July (Panel C) and October (Panel D).
Barren areas (e.g. the Sahara Desert) and any locations where overwintering is active for greater than 50% of the temporal record
are masked out. Barren areas are further masked using land cover data available from Li et al., 2018, part of the ESA Climate
Change Initiative - Land Cover led by UCLouvain (2017).

## 4.2 Validation for Canada

It is instructive to examine the accuracy of the ERA5 FWI System calculation when compared with station
observations, particularly given its intended use as a proxy to observed data. We perform a simple validation for
Canada, for which the FWI System was initially developed and calibrated. We used FWI values calculated from the
historical Environment and Climate Change Canada (ECCC) archive from 1979-2009, which represented the temporal
period of quality controlled data overlapping the ERA5 Reanalysis data (Natural Resources Canada - Canadian Forest
Service, Wildland Fire Information Systems, 2016).

For the validation, we considered three simple metrics: 1) Mean Absolute Error (mean($|X_1-X_0|$)); 2) Mean
Bias Error (MBE, mean($X_1-X_0$)); and 3) Spearman Rank Correlation (SRC, $\rho_s(X_0, X_1)$)., denoting $X_1$ as the FWI values
calculated from the ERA5 Reanalysis dataset and $X_0$ as the FWI values calculated using ECCC station data. Fig. 2
shows the spatial distribution of the three metrics with histograms of their values. Overall, citing the mean values with
the 5% and 95% percentiles (in square brackets) gives MAE=5.005049, 90%CI[1.58, 11.05], MBE=-3.6745624,
90%CI[-10.15, 0.26], and SRC=0.7764061, 90%CI[0.63, 0.88]. These results suggest that although there is a strong
correlation between the reanalysis and observed FWI values, the FWI values calculated from ERA5 exhibit a negative
bias, particularly in Alberta, Canada (see. Fig. 2b). Note, the higher density of stations in Alberta can be attributed to
a greater number of ECCC stations in the ECCC historical archive, including those from provincial weather station
networks including both Alberta Agriculture and Forestry and the Alberta Wildfire Management Branch.



An investigation into the source of the model bias is outside the scope of this paper. However, we note that
FWI values from the ERA5 reanalysis may be underestimated due to biases in wind speed and precipitation, as noted
in one recent study in Canada (Betts *et al.,* 2019). With respect to non-gauge corrected precipitation models, ERA5
performs well compared with other datasets (Beck *et al.,* 2019). Nevertheless, users of the dataset documented here
should be aware of any limitations in model bias/accuracy for their intended study area and period of interest.



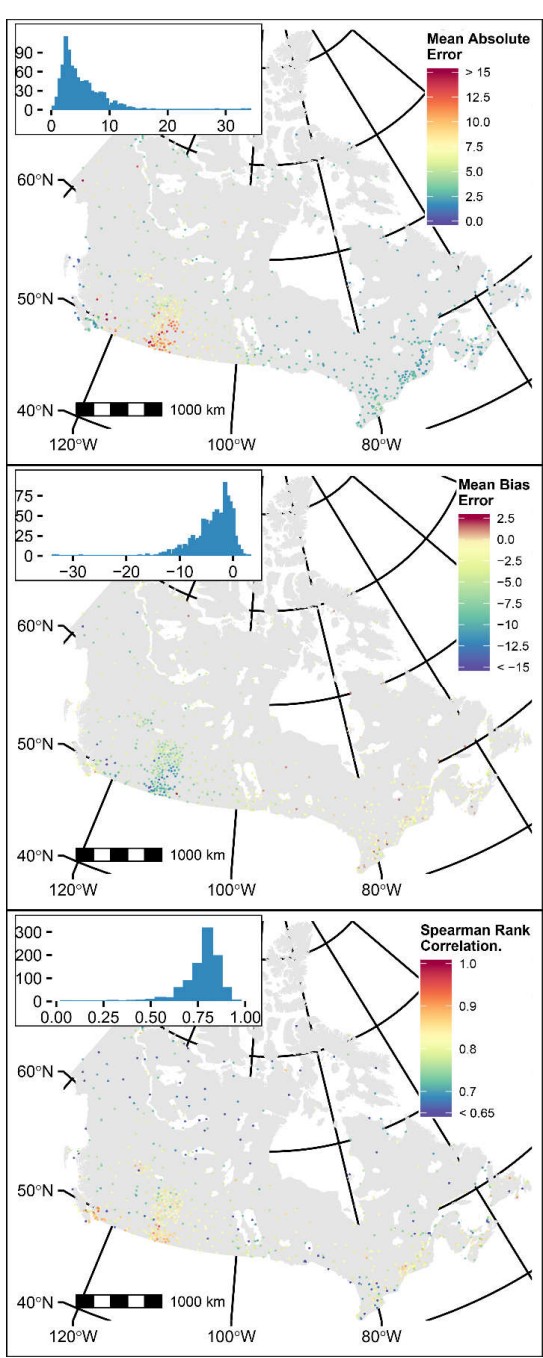

Fig. 2: Validation of FWI values calculated from the ERA5 reanalysis compared with observed FWI values calculated from ECCC station data for 1979-2009. Spatial distribution and histogram values are shown for the Mean Absolute Error (Panel A), Mean bias error (Panel B) and Spearman Rank Correlation (Panel C).





### 4.3 Effect of Overwintering the Drought Code

As discussed earlier, overwintering the Drought Code can modify the FWI System indices, particularly in areas with low overwinter precipitation and during spring (i.e. after snow melt, but before fuel moisture can be recharged from precipitation events). To explore this possibility further, we show differences between FWI calculations where the

process of overwintering the DC is performed and alternatively when the default DC start-up value (DC=15) is used. We focus on the case of calculated DC and FWI values for 2016 over North America. Fuel moisture preceding the Horse River fire (Fort McMurray, Alberta) in 2016 are widely considered to have been anomalously low due to low overwinter precipitation and severe fall drought conditions (Elmes *et al.,* 2018). Fig. 3 shows the day of year associated with fire season start-up (Panel A), the difference in DC values (overwintered vs default) at the corresponding start-

up day of year (Panel B), the corresponding difference in FWI values (Panel C) and the difference in spread day events for 2016 between the overwintered and default calculations (Panel D). These results show that even a modest difference in FWI values at start-up can lead to important differences in the number and spatial distribution of fire spread days between the overwintered and default calculations; in general, the greater number of fire spread days associated with the overwintered calculation may therefore account for sizeable differences in modelling area burned

where DC, BUI or FWI metrics are used as explanatory variables. Note that the results for other years (not shown) show a similar spatial pattern to the 2016 results.

We also found (not shown) that central and eastern Siberia displayed differences between the two calculations, most likely due to the low overwinter precipitation that occurs there (Stocks *et al.,* 1996). In general, we found that regions where overwintering leads to drier fuel moisture conditions correspond to areas of low overwinter

precipitation and were largely confined to western North America and parts of Eurasia. For regions where overwintering is likely to have an effect on spring fuel conditions, we therefore recommend using the version of the FWI calculation that overwinters the drought code.


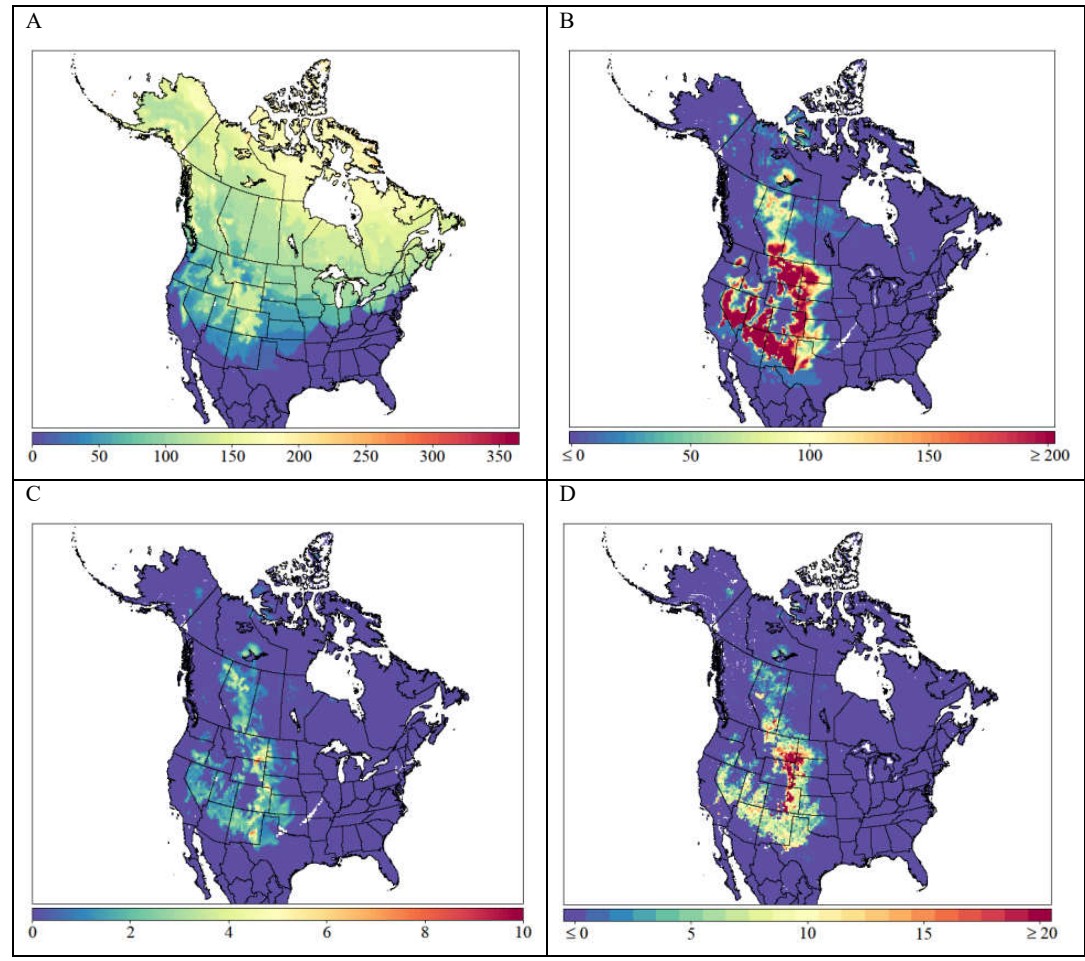

Fig. 3: Differences in FWI calculation using the default DC start-up value versus the overwintered DC start-up value for North America in 2016. Start-up day of year for FWI calculation based on longest period satisfying the meteorological fire season condition given by Wotton and Flannigan (1993) (Panel A). The difference between overwintered DC and default DC start-up values (i.e. DC=15) at the day of year given by Panel A (Panel B). The difference in FWI values corresponding to Panel B (Panel C). The corresponding difference in fire spread days (defined as FWI > 19 as per Podur and Wotton (2011)) (Panel D).


**5 Data Availability**


The FWI System indices calculated using both procedures (i.e. default and overwintered start-up values of the DC) can be downloaded from Zenodo as annual NetCDF files of daily values from https://doi.org/10.5281/zenodo.3626193 (McElhinny et al., 2020).



## 6 Conclusions

The Global Fire Weather Indices dataset developed from the ECMWF ERA5-HRS Reanalysis product is a publicly available global dataset that presents seven key variables representing fuel moisture (FFMC, DMC, DC) and potential fire behavior (ISI, BUI, FWI, and DSR). The dataset covers a period of 1979 to 2018 and accounts for the procedures of using the default DC or alternatively the overwintered DC to calculate fire behavior at fire season start-up. This dataset shows that there can be a significant difference in DC (and therefore also BUI, FWI and DSR) values,

particularly at the beginning of the fire season, depending on which procedure is employed, suggesting that fire danger in some regions may be more severe than what is predicted by the default DC.

The FWI Index calculated from the ECMWF data shows generally strong agreement with calculations based on Canadian weather station data (mean Spearman correlation of 0.77, mean absolute error of 5.0, and mean bias of -3.7). However, there are several caveats that are important to consider for users of the data. First, it is important to

note that the assumptions made for the overwintering process include that the carry-over fraction from the previous season's fall moisture (a) is always 1 and the coefficient for effectiveness of winter precipitation in recharging moisture reserves in the spring (b) is always 0.75. In reality these coefficients would vary spatially and temporally to reflect variations in topography as well as weather/climate. Second, as reanalyses represent modeled data, there are biases associated with model and/or data uncertainty. For example ERA5 has been shown to exhibit a negative daytime wind

speed bias in the Canadian Prairies (Betts *et al*., 2019). Lastly, although the resolution of the produced dataset is considered fine in relation to other reanalysis products (e.g. ERA-Interim), there may still be unresolved fine scale variations in fire behavior indices due to topographic or microclimatic variations. Regardless of these caveats, this dataset provides historical fire weather and potential fire behavior data that users should find useful for several research applications including calibration of FWI-based fire prediction models, historical relationships between fire weather

and fire danger at regional to global scales, baseline data for future fire danger projections under climate change scenarios, and analysis of regional or global trends in fire weather or behavior.

## Acknowledgements

The authors would like to thank Richard Carr, Denyse Dawe and Xinli Cai for their expert guidance.


## Appendix A

UTC time zone adjustments

| UTC Time Zone | Changed to |
|---|---|
| UTC-12:00 | UTC-11:00 |
| UTC-09:30 | UTC-09:00 |
| UTC-04:30 | UTC-04:00 |
| UTC-03:30 | UTC-04:00 |
| UTC+03:30 | UTC+04:00 |
| UTC+04:30 | UTC+05:00 |



| UTC+05:30 | UTC+06:00 |
|---|---|
| UTC+05:45 | UTC+06:00 |
| UTC+06:30 | UTC+07:00 |
| UTC+08:45 | UTC+09:00 |
| UTC+09:30 | UTC+09:00 |
| UTC+10:30 | UTC+11:00 |
| UTC+12:45 | UTC+12:00 |
| UTC+13:00 | UTC+12:00 |
| UTC+14:00 | UTC+12:00 |

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
