# Peer review of "A high-resolution reanalysis of global fire weather from 1979 to 2018 – Overwintering the Drought Code"

_Earth System Science Data, 2019_

## Referee Comment (RC1) · Anonymous Referee #1 · 22 Apr 2020

General Comments In this manuscript, the authors present a global high-resolution Fire Weather Index driven by data from ERA-5-HRS reanalysis. This work definitely adds value to the wildfire research field and it paves the road for further analysis and more studies. The method is adequate to the objectives and is well presented in the text. Some analysis could be added in order to improve the manuscript and make it a more robust work.

Specific Comments 1- In the paper, authors state that regional adjustments for the carry-over fraction from the previous season's fall moisture and the coefficient for the effectiveness of winter precipitation in recharging moisture reserves in the spring are

necessary when calculating the overwintered DC. The authors state that "As noted by Lawson and Armitage (2008) and Anderson and Otway (2003), the overwintered DC is most accurately represented when regional conditions are analyzed and the coefficients of the wDC function are adjusted accordingly. However, the ERA5 dataset did not contain information that allowed us to vary these coefficients and thus we chose the default values." How sensitive is the dataset to those variables a and b? (lines 223-229)

2- As been discussed in the paper, Reanalysis products have biases. The bias can transfer to the newly calculated products. To show the robustness of the proposed dataset, I think the validation should be repeated and shown for a few regions prone to wildfires like the western United States or Australia.

Technical corrections 1- The quality of Figure 2 should be improved. 2- Is this a continuing product? If it is, the authors should mention that in the manuscript.

---

## Referee Comment (RC2) · Anonymous Referee #2 · 28 Apr 2020

1. General considerations - This paper is easy to follow, provides a new method to calculate FWI using ERA5HRS data from 1979 to 2018 and uses two techniques to evaluate start-up value of the DC. Any efforts to reduce and anticipate damage from forest fires are welcome. - An important conclusion of the paper is that the dataset obtained shows some important differences in DC values depending on the procedure that the authors use. They conclude that the consequences of a forest fire may be worse in some regions compare with other predictions using default values of DC.

2. Global revisions to improve the paper - The data repository presents different spatial resolutions according to the years. Authors would need to display information about the

spatial resolution used in the repository's raster files and whether this spatial resolution depend on geographic location or not. - Is it possible to complete the data repository with the intermediate calculations or variables performed?

3. Particular revisions to improve the paper - Fig.1, Fig.2 and Fig.3 use a reference system, probably geographic coordinate system over WGS84 to show the maps. It is necessary to indicate the reference system used in all maps. It would be highly recommended to indicate throughout the paper (for example in the footnote under figures), the reference system used. - Fig. 3 shows a map for North America in 2016, but we do not know the reference system and no grid appears. - In section 3.3, the authors describe that R-cffdrs package is used for calculating FWI Systems outputs. It is very important to show the version of the packages used. The versions of the packages in R are necessary to reproduce the calculations the authors made. - There are several reported examples that using different versions of R packages produces different results in calculations. To improve reproducibility, I recommend the use of R packages such as the Git package. If this is not possible, the authors must show the list of all the packages used as well as the dependency tree, together with the version of R used. - Section 4.2 and Fig.2 represent validation for Canada. Some graphics appear in figures (upper left corner), but It would be very interesting to know if the represented histograms fit some known probability density function and what function might be. - In section 4.3, the authors present the specific statistical study for 2016 in North America (FWI index). It would be necessary to extend this study for several years, to see if the observed differences depend on the place or also depend on the time variable, showing a larger geostatistical study using time and position.

---

## Author Comment (AC2) · 21 Jun 2020

We thank the reviewer for their useful comments and provide our response here.

Original reviewer comment: 2. Global revisions to improve the paper - The data repository presents different spatial resolutions according to the years. Authors would need to display information about the spatial resolution used in the repository's raster files and whether this spatial resolution depend on geographic location or not. - Is it possible to complete the data repository with the intermediate calculations or variables performed?

[Figure]

Response: We are not sure what the referee is referring to here and would require further clarification. The spatial resolution of the FWI product is the same for all years since it is based on the input variables from ERA5 output at 0.25 degrees globally. This information is already given in lines 106-108 of the manuscript.

Original reviewer comment: 3. Particular revisions to improve the paper - Fig.1, Fig.2 and Fig.3 use a reference system, probably geographic coordinate system over WGS84 to show the maps. It is necessary to indicate the reference system used in all maps. It would be highly recommended to indicate throughout the paper (for example in the footnote under figures), the reference system used. - Fig. 3 shows a map for North America in 2016, but we do not know the reference system and no grid appears.

Response: We thank the referee for pointing this out and now have added the reference systems used for the maps in the revised paper for captions for Figs 1, 2 and 3.

Original reviewer comment: - In section 3.3, the authors describe that R-cffdrs package is used for calculating FWI Systems outputs. It is very important to show the version of the packages used. The versions of the packages in R are necessary to reproduce the calculations the authors made. - There are several reported examples that using different versions of R packages produces different results in calculations. To improve reproducibility, I recommend the use of R packages such as the Git package. If this is not possible, the authors must show the list of all the packages used as well as the dependency tree, together with the version of R used.

Response: We have identified the packages and version numbers used which we now include as a table in the appendix of the paper. We do not believe a dependency tree is necessary since such trees are not common in the literature and the moreover, the version numbers are sufficient for reproducibility of the results. We have therefore added a table of all packages and version numbers as a new Appendix B in the revised paper. We have also added a note to the manuscript (lines 198-199) to refer to this new table.

[Figure]

Original reviewer comment: - Section 4.2 and Fig.2 represent validation for Canada. Some graphics appear in figures (upper left corner), but It would be very interesting to know if the represented histograms fit some known probability density function and what function might be.

Response: The histograms shown in Fig. 2 represent the corresponding metrics. We are not sure what physical interpretation could be given to a fit of a known probability density function. Instead, it would seem to be more meaningful to fit distributions to the FWI values derived from stations and from the ERA5 reanalysis and to see if they belong to the same distribution or family of distributions. However, this is outside the scope of the present study which was to present an overwintered global calculation of the FWI System indices and present a simple validation. It may be of interest in further studies where, for example, such data is used to calibrate FWI projections under climate change using a parametric bias correction approach.

Original reviewer comment: - In section 4.3, the authors present the specific statistical study for 2016 in North America (FWI index). It would be necessary to extend this study for several years, to see if the observed differences depend on the place or also depend on the time variable, showing a larger geostatistical study using time and position.

Response: We agree that this would be an interesting extension of the present work. However a spatiotemporal study of the effects of the overwintering procedure on drought codes it outside the scope of this paper is left for future work. Here we sought to highlight a single year that demonstrated that significant differences between the default calculation with and without overwintering may occur.

---

## Author Response (AR1)

To the editor,

We thank the reviewers for their useful comments and provide below our responses to each comment.

Best regards,
Megan McElhinny, Justin F. Beckers, Chelene Hanes, Mike Flannigan, and Piyush Jain

**Anonymous Referee #1**

General Comments In this manuscript, the authors present a global high-resolution Fire Weather Index driven by data from ERA-5-HRS reanalysis. This work definitely adds value to the wildfire research field and it paves the road for further analysis and more studies. The method is adequate to the objectives and is well presented in the text. Some analysis could be added in order to improve the manuscript and make it a more robust work.

Specific Comments
1- In the paper, authors state that regional adjustments for the carry-over fraction from the previous season's fall moisture and the coefficient for the effectiveness of winter precipitation in recharging moisture reserves in the spring are necessary when calculating the overwintered DC. The authors state that "As noted by Lawson and Armitage (2008) and Anderson and Otway (2003), the overwintered DC is most accurately represented when regional conditions are analyzed and the coefficients of the wDC function are adjusted accordingly. However, the ERA5 dataset did not contain information that allowed us to vary these coefficients and thus we chose the default values." How sensitive is the dataset to those variables a and b? (lines 223-229)

Response: The *a* constant is only used to estimate the fall DC value if it was not measured i.e. the fire management agency turned off the weather station prior to ground freeze-up. In this case using the ERA-5 dataset the fall DC value is always known so using a value of 1 is most appropriate for *a*, and it should not be considered an adjustable parameter here (see Lawson and Armitage, 2008). Variation in the *b* constant will affect the start-up value of the DC to some extent, the bigger factor being the use of a value above the default (15) for which the total overwinter precipitation is most important to that calculation. It is also important to note that substantial spring or early summer precipitation will reduce the DC back down to 0 regardless of the starting value. Therefore, the analyses is more sensitive to precipitation inputs after the fire season starts. In any case, the *b* coefficient should be determined regionally based on soil conditions as well as annual variability of weather conditions. Typically, this requires in situ

measurements that are not possible at a continental or global scale. We therefore decided to use only the default value for the *b* coefficient of 0.75 which covers the most general situation.

2- As been discussed in the paper, Reanalysis products have biases. The bias can transfer to the newly calculated products. To show the robustness of the proposed dataset, I think the validation should be repeated and shown for a few regions prone to wildfires like the western United States or Australia.

Response: We agree that a global validation of the FWI product presented here would advantageous. However, we could not undertake such a validation at this time due to a lack of suitable input data. Although most countries provide meteorological station data, such data may not be quality controlled (eg. lack of homogenization) or may not include the required local 12 noon observations of all required variables for the FWI calculation. For example, weather data from the Australian Bureau of meteorology (http://www.bom.gov.au/climate/dwo/) includes daily weather at 9am, which does not correspond to the 12pm LST times of the FWI calculation. It should be noted that Tsinko et al. 2018 conclude that using raw over homogenized station data can lead to sizeable errors in the calculation of FWI. The amount of work required to collate and quality control the data necessary is outside the scope of this project. We therefore only validated for the FWI reanalysis for Canada because we had access to the required FWI input variables that had been quality controlled by Environment and Climate Change Canada (ECCC) for the period of the validation. We further note that only Northern latitudes or mountainous areas are expected to be affected by the overwintering procedure. For this reason the validation over Canada is important for our FWI calculation, particularly as overwintering may a moderate influence on the spring start up DC code in some regions in Canada.

Tsinko, Y., Bakhshaii, A., Johnson, E. A., & Martin, Y. E. (2018). Comparisons of fire weather indices using Canadian raw and homogenized weather data. *Agricultural and Forest Meteorology*, *262*, 110-119.

Technical corrections
    1- The quality of Figure 2 should be improved.

Response: A high resolution PDF version of Fig 2 will be provided to the journal for the final version of the manuscript.

    2- Is this a continuing product? If it is, the authors should mention that in the manuscript.

Response: This is not a continuing product and we have added a note to the text accordingly (Line 328 of revised manuscript).

Anonymous Referee #2

1. General considerations - This paper is easy to follow, provides a new method to calculate FWI using ERA5HRS data from 1979 to 2018 and uses two techniques to evaluate start-up value of the DC. Any efforts to reduce and anticipate damage from forest fires are welcome. - An important conclusion of the paper is that the dataset obtained shows some important differences in DC values depending on the procedure that the authors use. They conclude that the consequences of a forest fire may be worse in some regions compare with other predictions using default values of DC.

2. Global revisions to improve the paper - The data repository presents different spatial resolutions according to the years. Authors would need to display information about the spatial resolution used in the repository's raster files and whether this spatial resolution depend on geographic location or not. - Is it possible to complete the data repository with the intermediate calculations or variables performed?

Response: We are not sure what the referee is referring to here and would require further clarification. The spatial resolution of the FWI product is the same for all years since it is based on the input variables from ERA5 output at 0.25 degrees globally. This information is already given in lines 106-108 of the manuscript.

3. Particular revisions to improve the paper - Fig.1, Fig.2 and Fig.3 use a reference system, probably geographic coordinate system over WGS84 to show the maps. It is necessary to indicate the reference system used in all maps. It would be highly recommended to indicate throughout the paper (for example in the footnote under figures), the reference system used. - Fig. 3 shows a map for North America in 2016, but we do not know the reference system and no grid appears.

We thank the referee for pointing this out and now have added the reference systems used for the maps in the captions for Figs 1, 2 and 3.

- In section 3.3, the authors describe that R-cffdrs package is used for calculating FWI Systems outputs. It is very important to show the version of the packages used. The versions of the packages in R are necessary to reproduce the calculations the authors made. - There are several reported examples that using different versions of R packages produces different results in calculations. To improve reproducibility, I recommend the use of R packages such as the Git package. If this is not possible, the authors must show the list of all the packages used as well as the dependency tree, together with the version of R used.

We have identified the packages and version numbers used which we now include as a table in the appendix of the paper. We do not believe a dependency tree is necessary since such trees are not common in the literature and the moreover, the version numbers are sufficient for reproducibility of the results. We have therefore added the following table as a new Appendix B in the paper. We have also added a note to the manuscript (lines 198-199) to refer to this new table.

|  | Name | Version |
|---|---|---|
| 1 | raster | 2.9.5 |
| 2 | rgdal | 1.4.4 |
| 3 | gtools | 3.8.1 |
| 4 | ncdf4 | 1.16.1 |
| 5 | doParallel | 1.0.14 |
| 6 | abind | 1.4.5 |
| 7 | magclass | 4.107.0 |
| 8 | matrixStats | 0.54.0 |
| 9 | tseries | 0.10.47 |
| 10 | MASS | 7.3.51.4 |
| 11 | rgeos | 0.4.3 |
| 12 | cffdrs | 1.8.6 |
| 13 | devtools | 2.1.0 |
| 14 | rasterVis | 0.46 |
| 15 | accelerometry | 3.1.2 |
| 16 | ggplot2 | 3.2.0 |
| 17 | tdr | 0.13 |
| 18 | hydroGOF | 0.3.10 |

- Section 4.2 and Fig.2 represent validation for Canada. Some graphics appear in figures (upper left corner), but It would be very interesting to know if the represented histograms fit some known probability density function and what function might be.

Response: The histograms shown in Fig. 2 represent the corresponding metrics. We are not sure what physical interpretation could be given to a fit of a known probability density function. Instead, it would seem to be more meaningful to fit distributions to the FWI values derived from stations and from the ERA5 reanalysis and to see if they belong to the same distribution or family of distributions. However, this is outside the scope of the present study which was to present an overwintered global calculation of the FWI System indices and present a simple validation. It may be of interest in further studies where, for example, such data is used to calibrate FWI projections under climate change using a parametric bias correction approach.

- In section 4.3, the authors present the specific statistical study for 2016 in North America (FWI index). It would be necessary to extend this study for several years, to see if the observed differences depend on the place or also depend on the time variable, showing a larger geostatistical study using time and position.

Response: We agree that this would be an interesting extension of the present work. However a spatiotemporal study of the effects of the overwintering procedure on drought codes it outside the scope of this paper is left for future work. Here we sought to highlight a single year that demonstrated that significant differences between the default calculation with and without overwintering may occur.

[revised manuscript text omitted]